Comprehensive empirical evaluation of feature extractors in computer vision

http://orcid.org/0000-0003-3200-1609 ISIK Murat muratisik@ahievran.edu.tr
Computer Engineering, Faculty of Engineering and Architecture, Kirsehir Ahi Evran University , Kirsehir , Turkey
Alatas Bilal
Electronic publication date: 2024 Nov 4
Publication date: 2024
Volume: 10
Electronic Location ID: e2415
Received 2024 Apr 19; Accepted 2024 Sep 23
Copyright: © 2024 ISIK
Copyright year: 2024
Copyright holder: ISIK
License: This is an open access article distributed under the terms of the Creative Commons Attribution License, which permits unrestricted use, distribution, reproduction and adaptation in any medium and for any purpose provided that it is properly attributed. For attribution, the original author(s), title, publication source (PeerJ Computer Science) and either DOI or URL of the article must be cited.
License URL: https://creativecommons.org/licenses/by/4.0/

Keywords: Machine vision, Image processing, Feature extractors, Computer vision, Robustness in feature detection

Funding: The authors received no funding for this work.

==============================
Feature detection and matching are fundamental components in computer vision, underpinning a broad spectrum of applications. This study offers a comprehensive evaluation of traditional feature detections and descriptors, analyzing methods such as Scale Invariant Feature Transform (SIFT), Speeded-Up Robust Features (SURF), Binary Robust Independent Elementary Features (BRIEF), Oriented FAST and Rotated BRIEF (ORB), Binary Robust Invariant Scalable Keypoints (BRISK), KAZE, Accelerated KAZE (AKAZE), Fast Retina Keypoint (FREAK), Dense and Accurate Invariant Scalable descriptor for Yale (DAISY), Features from Accelerated Segment Test (FAST), and STAR. Each feature extractor was assessed based on its architectural design and complexity, focusing on how these factors influence computational efficiency and robustness under various transformations. Utilizing the Image Matching Challenge Photo Tourism 2020 dataset, which includes over 1.5 million images, the study identifies the FAST algorithm as the most efficient detector when paired with the ORB descriptor and Brute-Force (BF) matcher, offering the fastest feature extraction and matching process. ORB is notably effective on affine-transformed and brightened images, while AKAZE excels in conditions involving blurring, fisheye distortion, image rotation, and perspective distortions. Through more than 2 million comparisons, the study highlights the feature extractors that demonstrate superior resilience across various conditions, including rotation, scaling, blurring, brightening, affine transformations, perspective distortions, fisheye distortion, and salt-and-pepper noise.

Introduction

Feature extraction is a fundamental process in computer vision, serving as a critical component in various applications, including object recognition, texture recognition, image retrieval, image stitching, image alignment, image classification, reconstruction, navigation, and biometric systems (Jiang, 2009; Salau & Jain, 2019). In images, every pixel represents a piece of data; however, not all data contain useful information about the image characteristics. In computer vision, a “feature” refers to those data points that carry significant information, such as corners, blobs, edges, junctions, lines, and similar structures, which are related to the texture and shape of different regions of an image (Truong & Kim, 2016; Tareen & Saleem, 2018). Extracting a feature involves transforming the raw pixel values around these areas into numerical representations that can be processed by machine learning algorithms.

Feature detection and matching consist of three primary components: Detection, Description, and Matching.

Feature detection: Involves identifying distinctive points or regions within an image, known as keypoints or interest points, that are unique and can characterize the image’s content. For example, in an image of a building, keypoints might be detected at the corners of windows or edges of the roof, as these areas typically contain unique features.

Feature description: Once keypoints are detected, they are described using numerical values known as descriptors, which capture the local image information around each keypoint. This allows for the differentiation of one keypoint from another, even under varying conditions like changes in lighting or perspective.

Feature matching: Descriptors from different images are compared to find corresponding keypoints. For instance, in two images of the same building taken from different angles, the matching process would identify the same window corners in both images, enabling alignment or stitching of the images.

Given the foundational role of detection and description in feature extraction, the choice of appropriate detectors and descriptors is essential for accurately identifying keypoints and generating reliable descriptions. Various feature extraction techniques have been developed over the years, each offering unique strengths and addressing specific challenges in computer vision. Techniques such as Scale Invariant Feature Transform (SIFT) (Lowe, 2004), Speeded-Up Robust Features (SURF) (Tuytelaars & Gool, 2006), Binary Robust Independent Elementary Features (BRIEF) (Calonder et al., 2010), Oriented FAST and Rotated BRIEF (ORB) (Rublee et al., 2011), Binary Robust Invariant Scalable Keypoints (BRISK) (Leutenegger, Chli & Siegwart, 2011), KAZE (Alcantarilla, Bartoli & Davison, 2012), Accelerated KAZE (AKAZE) (Alcantarilla, Nuevo & Bartoli, 2013), Fast Retina Keypoint (FREAK) (Alahi, Ortiz & Vandergheynst, 2012), Dense and Accurate Invariant Scalable descriptor for Yale (DAISY) (Tola, Lepetit & Fua, 2009), Features from Accelerated Segment Test (FAST) (Rosten & Drummond, 2006), and STAR (Agrawal, Konolige & Blas, 2008) have been selected for this study due to their widespread adoption, proven effectiveness, and ability to address different aspects of feature matching, including scale and rotation invariance, computational efficiency, and robustness under challenging conditions. These traditional techniques have not only laid the groundwork but also remain integral to deep learning approaches, often serving as key components in pre-processing stages or benchmarks for evaluating advanced models.

To determine which algorithm excels in speed, robustness, and sensitivity to variations caused by rotation, affine transformation, perspective transformation, scaling, brightening, and blurring, our study systematically evaluated each algorithm across various transformations. The novelty of this study lies in its extensive and systematic evaluation of 10 fundamental feature extractors across a broad spectrum of conditions using a large-scale dataset. This article provides the most detailed benchmark to date for traditional feature extractors, highlighting their strengths and weaknesses across various scenarios. The findings advance the understanding of these algorithms’ performance and establish a valuable reference for future research and applications requiring robust and efficient feature extraction.

The remainder of this article is organized as follows: “Related Works” reviews related work in the field of feature extraction. “Methods” describes the methodology and experimental setup used in this study. “Results” and “Discussion” presents the results and discussion, highlighting key findings. “Conclusions” concludes the article and suggests directions for future research.

Related works

Tareen & Saleem (2018) conducted a comparative analysis of SIFT, SURF, KAZE, AKAZE, ORB, and BRISK, concluding that SIFT, SURF, and BRISK perform better on scaled images, while AKAZE is more effective on rotated images. Additionally, ORB and BRISK are better suited for affined images. Chien et al. (2016) reviewed SIFT, SURF, ORB, and AKAZE feature extractors for Monocular Visual Odometry using a ready-made benchmark dataset. Their results indicated that SURF offers the highest accuracy, while ORB is the most computationally efficient solution. AKAZE, on the other hand, demonstrated a balance between accuracy and computational efficiency.

Andersson & Reyna Marquez (2016) performed a comparative analysis of SIFT, KAZE, AKAZE, and ORB using a novel dataset of over 170 test images subjected to transformations such as scaling, rotation, and illumination adjustments. Their findings highlighted SIFT’s superior performance across all tests, while AKAZE showed comparable accuracy to KAZE with improved computational efficiency. Tareen & Raza (2023) conducted a quantitative comparison of 14 feature detectors, including SIFT, SURF, KAZE, AKAZE, ORB, BRISK, AGAST, MSER, MSD, GFTT, Harris Corner Detector-based GFTT, Harris Laplace Detector, and CenSurE. The evaluation utilized 10 image pairs derived from the original images through the addition of imperceptible elements such as dust, smoke, darkness, noise, motion blur, extreme affine transformations, JPEG compression, occlusions, shadows, and lighting effects, achieving varied results per derived image.

Aydin (2022) conducted an experimental study comparing the effectiveness of KAZE, SURF, and ORB descriptors in face identification tasks. The results revealed that KAZE outperformed the others in all tested scenarios. Bansal, Kumar & Kumar (2021) evaluated the performance of SIFT, SURF, and ORB feature descriptors against rotation, scaling, shearing, intensity variations, and noise using the Caltech-101 public dataset. Their findings suggest that a hybrid approach combining SIFT, SURF, and ORB yields optimal results.

Yusefi & Durdu (2020) developed a two-step algorithm involving keypoint identification and outlier removal, testing five local feature detection algorithms: SURF, SIFT, FAST, STAR, and ORB. Using a grayscale dataset with a resolution of 1,241 × 376 pixels, their experiments showed that FAST outperformed other detectors in visual odometry accuracy, while ORB and STAR offered better computational runtime at the cost of accuracy. Liu, Xu & Wang (2021) introduced handcrafted detectors, including FAST, BRISK, ORB, SURF, SIFT, and KAZE, and descriptors, including BRISK, FREAK, BRIEF, SURF, ORB, SIFT, and KAZE. They worked with two datasets—the publicly available natural light collection dataset and specialized multimodal data—achieving varied results depending on the dataset.

While this study focuses on traditional feature extractors and provides a comprehensive evaluation of their performance across various transformations, recent advancements in state-of-the-art (SOTA) deep learning techniques should be acknowledged. For example, Roy & Bhaduri (2023) enhanced the feature extraction process by integrating advanced components into the YOLO architecture, specifically for real-time damage detection in infrastructure. Similarly, Roy, Bose & Bhaduri (2022) discuss the implementation of YOLO v4, which excels in accuracy and speed, making it suitable for real-time object detection in complex environments. Hassan et al. (2022) proposed a 28-layer convolutional neural network (CNN) model for classifying infected cases, while Hassan et al. (2023) explored optimization strategies for computer vision tasks, particularly in medical image analysis. Despite the potential of deep learning solutions, challenges such as increased computational complexity, longer training times, potential overfitting, hardware dependency, and issues with model interpretability persist. The insights gained from this study highlight the strengths and limitations of traditional methods, which remain relevant in many computer vision tasks despite the growing prominence of deep learning.

Given the critical role of feature extraction and description in machine vision, researchers have consistently sought to refine and enhance these techniques. According to Google Scholar, several algorithms examined in this study have garnered over 85,000 citations, contributing to a substantial body of comparative literature. The novelty of this article lies in the evaluation and comparison of 10 fundamental feature extractors/descriptors using a consistent dataset, yielding quantitative comparison results. The discussion section systematically examines and elucidates the disparities between analogous studies and the present investigation.

Feature extractors

SIFT is a widely used method for object recognition (Chiu et al., 2013), which transforms image data into scale-invariant coordinates relative to local features. The algorithm consists of four major stages. The first stage, scale-space extrema detection, identifies potential interest points that are invariant to scale and orientation. The algorithm works by detecting keypoints at multiple scales using a Difference of Gaussian (DoG) function, selecting them based on stability measures. One or more orientations are then assigned to each keypoint location based on local image gradient directions. These gradients are measured at the selected scale within the region around each keypoint.

SURF is similar to SIFT but is designed to be faster and more efficient. It relies on integral images to reduce computation time, and keypoints are refined by computing the Hessian matrix at each point. The descriptor vector is calculated by assessing the Haar wavelet responses within the interest point neighborhood. An additional indexing step, based on the sign of the Laplacian, enhances the matching speed and robustness of the descriptor (Bay et al., 2008).

BRIEF, inspired by earlier works (Ozuysal et al., 2009; Lepetit & Fua, 2006), randomly selects pixel point pairs around a feature point. By comparing the grey values of these selected pairs, BRIEF produces a binary string as the feature description. It directly computes binary strings from image patches into a binary feature vector consisting solely of 1 s and 0 s. Each keypoint is thus described by a bit string, and BRIEF compares these strings using the Hamming distance, allowing for rapid computation. However, this method is highly sensitive to noise.

DAISY is a feature descriptor introduced as a reformulation of SIFT and Gradient Location-Orientation Histogram (GLOH) descriptors, enabling efficient computation at every pixel location while retaining the robustness of these methods. It relies on histograms of gradients, similar to SIFT and GLOH, but uses a circularly symmetrical weighting kernel (Tola, 2010; Tola, Lepetit & Fua, 2008). The Gaussian convolutions involved in DAISY computation allow for GPU implementation, leading to real-time or faster computation of descriptors across all image pixels (Tola, Lepetit & Fua, 2009).

ORB is a robust algorithm designed for real-time applications in computer vision. It combines the FAST algorithm for initial keypoint detection with the Harris corner response to enhance performance. ORB assigns orientations to keypoints to ensure rotation invariance and uses the BRIEF algorithm to generate binary descriptors. These descriptors are rotated according to the assigned orientation, providing stability against image rotations. ORB’s computational efficiency and capability in keypoint matching make it particularly valuable for tasks like image stitching and object recognition (Rublee et al., 2011).

BRISK is a prominent feature extractor known for its robustness and scalability. It introduces a binary descriptor to enhance computational efficiency, providing reliable and invariant keypoints across diverse image conditions. Key components include a scale space for achieving scale invariance, non-maximum suppression for identifying salient keypoints, and a well-defined sampling pattern around each keypoint. BRISK’s adaptability to different image scales and the binary nature of its descriptors contribute to its computational efficiency, facilitating fast keypoint matching between images. It is recognized for its resilience in challenging scenarios and is frequently cited in academic literature (Leutenegger, Chli & Siegwart, 2011).

KAZE is an advanced feature extractor designed to enhance the speed and robustness of keypoint detection and description. As an improvement over traditional methods like SIFT and SURF, KAZE incorporates nonlinear diffusion for image smoothing, which is particularly beneficial in noisy or textured images. The multiscale approach identifies keypoints across various image scales, while the nonlinear scale space-based descriptor integrates intensity and gradient information, contributing to the resilience and distinctiveness of the feature representation. KAZE is noted for its adaptability to varied image conditions and its ability to deliver invariant features amidst noise and illumination variations (Alcantarilla, Bartoli & Davison, 2012).

AKAZE is an evolution of the KAZE algorithm, specifically designed for efficient and robust keypoint detection and description. AKAZE improves computational speed while preserving the robustness of feature extraction. It employs nonlinear diffusion for image smoothing, proving effective in handling textured or noisy images, and uses a multiscale approach for detecting keypoints across diverse image scales. The descriptor, based on the nonlinear scale space, integrates intensity and gradient information, ensuring resilient and distinctive feature representation. AKAZE is known for its adaptability to varied image conditions, excelling in scenarios involving noise and illumination changes (Alcantarilla, Nuevo & Bartoli, 2013).

FREAK is a feature extractor designed for fast and efficient keypoint detection and description. Positioned as an alternative to traditional methods like SIFT and SURF, FREAK prioritizes computational efficiency while maintaining robustness. It uses binary descriptors to capture intensity differences between pairs of retina-like sampling points. The retinal sampling pattern enhances robustness against scale changes, and FREAK incorporates orientation assignment to ensure rotational invariance. FREAK is recognized for its effectiveness in scenarios demanding computational speed, making it particularly suitable for real-time applications (Alahi, Ortiz & Vandergheynst, 2012).

FAST is widely used in computer vision for its efficiency in real-time feature extraction. It identifies keypoints by comparing pixel intensities in a circular pattern around a central pixel, classifying a set of contiguous pixels as a corner based on predefined intensity thresholds (Rosten & Drummond, 2006). FAST is known for its computational speed, making it particularly suited for real-time applications. However, it lacks orientation information for keypoints, rendering it sensitive to image rotations. Despite this limitation, its rapid processing and effectiveness make it a valuable tool for various computer vision tasks.

The STAR feature detector, an evolved variant of CenSurE, was introduced by the OpenCV library. Utilizing the Laplacian of Gaussians (LoG), STAR uses two overlapping squares at a 45-degree difference, approximating a circular mask commonly used in feature detection methodologies (Agrawal, Konolige & Blas, 2008; Patel et al., 2014).

In the comparative analysis, it is essential to recognize that certain feature extractors serve unique roles as detectors, descriptors, or both. Therefore, the analysis will be categorized into Detection and Description. Table 1 provides a comprehensive overview of each feature extractor, detailing its role as a detector, descriptor, or both.

Table 1 Feature extractors.

Algorithm	Detector	Descriptor	
SIFT	X	X	
SURF	X	X	
ORB	X	X	
BRISK	X	X	
KAZE	X	X	
AKAZE	X	X	
BRIEF		X	
FREAK		X	
DAISY		X	
FAST	X		
STAR	X		

Methods

This study focused on evaluating traditional feature extractors in terms of their computational efficiency and robustness. Although no new model was introduced, each feature extractor was assessed based on its architectural design and complexity. A third-party dataset named Image Matching Challenge Photo Tourism 2020, containing over 1.5 million images from 10 distinct locales, was utilized for analysis and comparison (Jin et al., 2021). The images were obtained from the dataset’s publicly available repository and were used in accordance with the dataset’s license, which permits their use in academic publications. The dataset is accessible (IMCP, 2020; Işik, 2024).

From this extensive dataset, a subset of 4,282 images from distinct locales was randomly selected. These images served as the foundation for generating 196,972 variations, ensuring a comprehensive evaluation of the feature extractors. The variations included a wide range of transformations, such as affine transformation, blurring, Gaussian blurring, fisheye distortion, horizontal perspective, brightening, rotation, scaling, salt-and-pepper effects, and vertical perspective. Table 2 presents the quantities of both the original images and their transformed counterparts, resulting in a final dataset of 201,254 images used for the comparative analysis of feature extractors, which constitutes the primary focus of this study. Figure 1 illustrates the subset of images and one of their corresponding variations. The details of these variations are further explained in the Image Deriving Strategy section.

Table 2 Counts of images utilized in the study.

Transformations	Images of Taj Mahal	Images
palace of westminster	Images of grand place brussels	Images of Temple Nara Japan	Total images	
Original	1,312	983	1,083	904	4,282	
Rotation	10,496	7,864	8,664	7,232	34,256	
Scaling	7,872	5,898	6,498	5,424	25,692	
Blurring	5,248	3,932	4,332	3,616	17,128	
Blurring gaussian	5,248	3,932	4,332	3,616	17,128	
Brightening	5,248	3,932	4,332	3,616	17,128	
Affine transformation	5,248	3,932	4,332	3,616	17,128	
Fisheye	1,312	983	1,083	904	4,282	
Horizontal perspective	6,560	4,915	5,415	4,520	21,410	
Vertical perspective	6,560	4,915	5,415	4,520	21,410	
Salt and pepper	6,560	4,915	5,415	4,520	21,410	
Total comparison	61,664	46,201	50,901	42,488	201,254	

Figure 1 Examples of the subset and its variations.

(A) Original. (B) Affine Transformation. (C) Blurring. (D) Gaussian blurring. (E) Fisheye disorder. (F) Horizontal Perspective. (G) Brightening. (H) Rotation. (I) Scaling. (J) Salt and Pepper effects. (K) Vertical perspective.

Image deriving strategy

Various transformations were manually applied to the original images to compute the matching evaluation parameters, including the number of keypoints, the time required for description and detection, and the overall matching performance. For example, consider an original image subjected to a rotation transformation. The original image, with dimensions of 1,024 × 768 pixels, initially had 550 keypoints detected using ORB. After a 40-degree clockwise rotation, 498 keypoints were detected in the transformed image. The BF Matcher was employed to compare these keypoints, resulting in 450 successful matches in 0.3 s, corresponding to a matching accuracy of 81.8%. In contrast, k-Nearest Neighbors (kNN) Matcher achieved a higher accuracy of 88.7% but required 0.6 s for matching. The time taken for detecting and describing the keypoints in the original image was 0.20 s and 0.25 s, respectively, while it took 0.18 and 0.22 s for the rotated image. This process was systematically applied to all the algorithms under study, allowing for a comprehensive comparison of their performance across various transformations. The following transformations were applied:

Rotation: All original images underwent a clockwise rotation in 40-degree increments within the 0 to 360-degree range, resulting in eight variations per image. Rotation refers to the process of turning or pivoting the entire image around a specified point or axis. This transformation changes the orientation of the image without altering the shape of its individual elements.

Scaling: The image dataset underwent a systematic rescaling process, commencing at 20% and incrementing by 50% up to 300%, creating six variations per image. Rescaling involves adjusting an image’s size or dimensions while preserving its original proportions. A 100% scale maintains the original size, a 50% scale reduces both width and height by half, and a 300% scale triples both dimensions.

Blurring: In image processing, “blurring” reduces sharpness and detail, resulting in a smoother appearance. The extent of blurring is determined by parameters such as the size of the blur kernel. Exclusively odd-numbered kernel structures were employed, aligning with the Gaussian blur technique’s inherent compatibility with odd-numbered kernel sizes. The original images were subjected to a blurring process using kernel structures ranging from 1 × 1 to 11 × 11. Notable blur methods include Gaussian, mean, median, and bilateral filtering, but this study specifically utilized the Gaussian blur method with a sigma value of 0 and kernels ranging from 3× to 15×, as well as the averaging blur method with kernels ranging from 4× to 13×. From the blurring procedure, 8 variations were created for each image.

Brightening: The procedure of “brightening” involves increasing the overall luminosity or brightness of an image to enhance visibility, highlight details, or address underexposure. Conversely, “darkening” reduces luminosity or brightness, used to decrease visibility or correct overexposure. In this study, gamma correction within the range of 0.4 to 1.6, using incremental steps of 0.4, was employed, resulting in four variations per image. Gamma correction modifies the brightness and contrast of an image, with a gamma value less than 1.0 darkening the image and a value greater than 1.0 lightening it. A gamma value of 1.0 has no effect, maintaining the original pixels.

Affine transformation: In image processing, affine transformation involves linear transformations such as translation, rotation, scaling, and shearing while preserving parallel lines. To achieve this transformation, a matrix multiplication (linear transformation) is followed by a vector addition (translation) (OpenCV, 2024). In this study, images were affined in 5-degree increments within the 15 to 30-degree range, generating four variations per image.

Fisheye distorted: Fisheye distortion, prevalent in imagery captured through fisheye lenses with a field of view often surpassing 180 degrees, introduces significant geometric distortions, particularly curving straight lines towards the image periphery. Two forms of fisheye distortion, barrel, and pincushion, exist. In this study, barrel distortion was applied to the entire image dataset, creating one variation per image.

Horizontal and vertical perspective: These transformations alter the vertical or horizontal perspective of an image to correct or manipulate perceived depth and orientation. Vertical perspective transformations preserve the parallelism of horizontal edges, while horizontal perspective transformations maintain the parallelism of vertical edges. Both horizontal and vertical perspectives were applied individually to all images in the dataset, ranging from 5° to 25° with a 5° increment, leading to five variations for each type of perspective distortion per image.

Salt-and-pepper noise: This noise model was added to all original images with a density ranging from 0.1 to 0.26, with 0.6 steps, resulting in five variations per image. Salt-and-pepper noise involves the introduction of black and white dots, simulating data corruption and challenging the feature extractors’ robustness.

Experimental setup

Figure 2 illustrates the schematic representation of the proposed methodology. In the initial phase, the transformations were applied to all images. Subsequently, the post-transformation images underwent systematic comparative analysis with their respective original counterparts using feature descriptors and employing two distinct matching methods. The experiments in this study were conducted on a computer equipped with an Intel® Core™ i7-7700 CPU (3.6 GHz with 4.2 GHz max Turbo Frequency), 8 MB cache, 16 GB of installed memory (RAM), and a 64-bit operating system. Additionally, each implementation in the study was performed using Python with the OpenCV library version 4.5.

Figure 2 Block diagram of the proposed method.

Matching strategy

After extracting keypoints and their corresponding descriptors from two images, the primary objective is to establish correspondences among these keypoints. Each keypoint is intricately associated with a descriptor, providing a numerical representation that characterizes the local image neighborhood surrounding the keypoint. Subsequently, the BF Matcher systematically conducts pairwise comparisons involving all descriptors extracted from the first image with those obtained from the second image. For each pair of descriptors, a distance metric—often the Euclidean distance—is computed to quantify the degree of similarity or dissimilarity within the respective feature vectors. A smaller distance corresponds to a higher level of similarity. The matcher generates a compilation of potential matches, with each entry comprising a pair of keypoints accompanied by an associated distance value.

In this study, two types of matchers were established. The first one is the BF matcher (also known as the normal matcher) with cross-checking, designed to obtain only the best matches. The second is a kNN matcher (with ‘k’ set to 2), where the BF matcher returns two best matches instead of only the best match. For the kNN matcher, the ratio test proposed by Lowe (2004) was employed as a criterion to filter out unreliable matches. The ratio test involves comparing the distance of the best match with the distance of the second-best match. If the ratio of the distances falls below a certain threshold of 0.75, the match is considered acceptable. The rationale behind the ratio test is that a significantly better best match compared to the second-best match indicates reliability. Conversely, if the distances are similar, it suggests ambiguity, and the match may be discarded.

For binary string-based descriptors such as ORB, BRIEF, BRISK, and AKAZE, the Hamming distance was employed as the distance metric. For floating-point-based descriptors such as SIFT, SURF, FREAK, KAZE, and DAISY, the Norm_L2 distance metric was used.

Results

The study utilized 4,282 images, resulting in a total of 196,972 variations to conduct a comprehensive comparative analysis of feature extractors. The primary objective was to evaluate the robustness of each feature extractor against a range of transformations, including affine transformation, blurring, Gaussian blurring, fisheye distortion, horizontal perspective, brightening, rotation, scaling, salt-and-pepper noise, and vertical perspective. The feature extractors evaluated in this study serve as baseline methods for various computer vision tasks. These methods were benchmarked against a standardized dataset, and their performance was compared across multiple transformations. While the study does not propose a new model, it provides a clear benchmark for each extractor’s strengths and weaknesses, offering valuable insights for practitioners selecting the appropriate feature extraction method for specific applications.

The matching process revealed instances where no matches were found when attempting to correspond features extracted using certain detectors and descriptors. Table 3 presents the aggregate count of absence of matches observed during the feature matching process. Table 4 offers a detailed analysis of the computational time performance of the feature detection and description algorithms, providing insights into the average detection and descriptor times, the standard deviation of these times, as well as the minimum and maximum values observed across the algorithms, excluding zero values. Table 5 presents the average number of matched features and the corresponding matcher time for both the normal matching method and the kNN matching method, along with the standard deviation of the matching times. It additionally provides the average percentage of successful matches for the two matchers.

Table 3 Absence of matches in feature matching process.

Detector	ORB	AKAZE	BRISK	KAZE	SIFT	SURF	STAR	STAR	STAR	FAST	
Descriptor	ORB	AKAZE	BRISK	KAZE	SIFT	SURF	BRIEF	FREAK	DAISY	ORB	
Affine transformation	1	12	0	0	0	0	56	56	56	0	
Blurring	34	29	48	1	3	1	112	118	112	100	
Gaussian blurring	20	16	27	0	0	0	69	70	69	14	
Fisheye	5	8	3	1	0	1	20	20	20	5	
Horizontal perspective	1	17	0	0	0	0	67	68	67	0	
Brightening	1	14	1	3	0	0	69	69	69	0	
Rotation	5	28	0	0	0	0	105	106	105	1	
Scalation	6	23	2	0	0	0	84	84	84	5	
Salt pepper	0	16	0	0	0	0	65	65	65	0	
Vertical perspective	1	16	0	0	0	0	67	69	68	0	
Total	74	179	81	5	3	2	714	725	715	125	

Table 4 Time duration of feature extracting process.

Detector	Descriptor	Average detector time for per keypoint (us)	Average descriptor time for per keypoint (us)	Min of detector time (sn)	Max of detector time (sn)	Min of descriptor time (sn)	Max of descriptor time (sn)	Standard deviation of detector time	Standard deviation of descriptor time	
ORB	ORB	1.2316	2.7737	0.1369	66.6323	0.0890	60.1282	0.396675	0.308078	
AKAZE	AKAZE	45.9090	75.8527	0.9455	1,276.0176	0.8108	1,276.0176	0.379389	0.291862	
BRISK	BRISK	12.3943	22.4021	0.2875	1,585.6814	0.0276	1,585.6814	0.443202	0.325694	
KAZE	KAZE	247.1228	319.5537	3.8335	7,782.6029	4.0549	7,782.6029	0.677204	0.496662	
SIFT	SIFT	38.9270	61.8236	1.1138	1,642.4004	0.7949	1,642.4004	0.381064	0.295739	
SURF	SURF	15.6335	69.6294	0.5286	1,049.9460	0.1688	1,049.9460	0.379766	0.309790	
STAR	BRIEF	39.7323	7.9004	0.1295	1,178.9323	0.0258	1,178.9323	0.385898	0.308341	
STAR	FREAK	39.9085	12.6464	0.1283	1,164.1974	0.0315	1,164.1974	0.385933	0.307899	
STAR	DAISY	38.8277	385.7660	0.1372	1,171.927	3.3826	1,171.9273	0.386111	0.293872	
FAST	ORB	0.2884	2.7646	0.0208	60.4136	0.0948	60.4136	0.396905	0.308151	

Table 5 Keypoints matching performance.

Detector	Descriptor	Keypoints count
(Average)	BF/kNN matched keypoint count
(Average)	Average BF matcher time for per feature (us)	Average kNN matcher time for per feature (us)	Average successful BF matches (%)	Average successful kNN matches (%)	Standard deviation of BF matcher time	Standard deviation of kNN matcher time	
ORB	ORB	5,490	3,035/1,580	2.7646	63.4001	55.2842	28.7769	0.282560	0.282302	
AKAZE	AKAZE	1,963	1,297/935	19.9852	28.7736	66.0797	47.6500	0.287316	0.286834	
BRISK	BRISK	4,838	2,615/1,535	60.6235	103.8791	54.0643	31.7397	0.349487	0.346301	
KAZE	KAZE	2,414	1,432/991	45.4907	67.0615	59.3276	41.0783	0.276005	0.275680	
SIFT	SIFT	4,337	2,298/1,628	129.6009	184.1318	52.9960	37.5316	0.339930	0.339640	
SURF	SURF	4,484	2,388/1,452	101.2370	168.8419	53.2522	32.3772	0.318859	0.323384	
STAR	BRIEF	772	399/218	7.4322	14.5048	51.7263	28.3523	0.303898	0.303699	
STAR	FREAK	765	407/140	19.0981	56.5929	53.2577	18.3154	0.299956	0.299751	
STAR	DAISY	772	328/187	62.7863	110.8715	42.5569	24.2885	0.290838	0.290507	
FAST	ORB	9,801	4,654/1,907	82.7081	203.2720	47.4837	19.4607	0.492367	0.488000	

For Figs. 3 to 9, the x-axis represents the feature extractor applied to the images, while the y-axis indicates the matching accuracy percentage of the feature extractors. Figure 3 illustrates the matching performance of the feature extractors when the same image is subjected to Affine Transformation. The figure demonstrates the effectiveness of each feature extractor in preserving accurate matches under affine transformations, offering valuable insights into their robustness and suitability for such conditions.

Figure 3 Matching achievement of feature extractors against affine transformation.

Figure 4 Matching achievement of feature extractors against blurring and Gaussian blurring.

Figure 5 Matching achievement of feature extractors against fisheye effect.

Figure 6 Matching achievement of feature extractors against vertical and horizontal perspective.

Figure 7 Matching achievement of feature extractors against brightening and salt-and-pepper noise.

Figure 8 Matching achievement of feature extractors against rotation.

Figure 9 Matching achievement of feature extractors against scaling.

Figure 4 depicts the matching performance of the feature extractors under the influence of blurring. It provides a visual representation of how each feature extractor performs in maintaining accurate matches when images are subjected to blurring. Figure 5 shows the matching performance of the feature extractors when images are confronted with the fisheye effect. Through visual representation, it elucidates how each feature extractor performs in preserving accurate matches amidst fisheye distortion.

Figure 6 illustrates the matching performance of feature extractors when images are subjected to vertical and horizontal perspective transformations. The graph offers insights into the capability of each feature extractor to manage changes in perspective orientation. Figure 7 depicts the matching performance of the feature extractors under varying degrees of image brightening and in the presence of salt-and-pepper noise. The graph provides valuable insights into the performance of each feature extractor as image brightness changes, as well as its effectiveness in the presence of salt-and-pepper noise.

Figure 8 illustrates the matching performance of the feature extractors across different degrees of image rotation. The graph delineates how each feature extractor performs in maintaining accurate matches amidst image rotation. Figure 9 depicts the matching efficacy of the feature extractors across varying degrees of image scaling. The graph visually elucidates their ability to produce reliable matches in the presence of scaling transformations.

Discussion

In the evolving field of image processing, the capability of feature extractors to maintain accuracy under varying conditions remains a cornerstone for evaluating their performance. This study aims to analyze well-established conventional feature extractors, including SIFT, SURF, BRIEF, ORB, BRISK, KAZE, AKAZE, FREAK, DAISY, FAST, and STAR, under the influence of rotation, scaling, blurring, brightening, affine transformation, horizontal/vertical perspective transformation, fisheye distortion, and salt-and-pepper noise.

Earlier studies mainly tested algorithms with a small set of images, which often caused variations in timing assessments and matching results. These differences were mostly due to the unique features of the images used in the tests.

Table 6 summarizes previous research and points out the differences from this study. Numerous studies have compared algorithms, but many of these evaluations are conducted on a limited number of images, leading to variability in time-related evaluations or matching procedures due to the nature of the images. For example, if a significant portion of an image consists of uniform color tones, such as the sky, some algorithms may inherently struggle to produce successful matching results. Similarly, studies involving a large number of images often present only a few feature extractors’ comparison results. However, in this study, a total of 201,254 comparison operations were conducted for each feature extractor, involving the matching of 4,282 original images and 196,972 transformed images derived from the originals. With over 2 million comparison operations in total, conclusions were drawn regarding which feature extractor was successful under specific variations and which one was faster than the others.

Table 6 Comparison with previous studies.

Study	Explanation	Differences	
Tareen & Saleem (2018)	In the study, six feature extractors were used for analysis and comparison.	Their study was limited to a smaller dataset and fewer transformations, which may not fully capture the diversity of real-world conditions. In contrast, our study uses a significantly larger dataset and examines a broader range of transformations.	
Chien et al. (2016)	Four feature extractors were compared for a specific application.	Their study focused on specific use cases within monocular visual odometry and did not encompass all feature extractors discussed in this article. Additionally, they did not investigate operations such as blurring, rotating, or scaling, which our study includes.	
Andersson & Reyna Marquez (2016)	A comparative analysis of four algorithms was presented, namely SIFT, KAZE, AKAZE, and ORB.	While the study provided insights into accuracy and computational efficiency, it did not encompass all feature extractors discussed in this article. Additionally, it did not investigate the impact of operations such as blurring, distorting, and scaling, which our study comprehensively covers.	
Tareen & Raza (2023)	A comparative analysis of 14 algorithms was presented.	Their study was restricted by the limited scope of transformations and dataset size. Our study, however, involves over 2 million comparisons, covering a wider array of transformations, providing more comprehensive and statistically robust results.	
Aydin (2022)	KAZE, SURF, and ORB extractors were compared on the Face 94 dataset.	The investigation predominantly concentrated on facial images, with a selective comparison of only three feature extractors. Notably, the study omitted the assessment of variants such as blurring, smoothing, and alterations in perspective, which are covered extensively in our study.	
Bansal, Kumar & Kumar (2021)	SIFT, SURF, and ORB feature descriptors were studied using the Caltech-101 public dataset.	The study focused on only three feature extractors and did not address brightening, affine transformations, or perspective transformations. Our study covers 10 feature extractors across a broader set of image transformations, providing a more holistic evaluation.	
Liu, Xu & Wang (2021)	Six detectors and seven descriptors were compared on two different datasets.	The study included only five types of image transformations (compression, illumination, image blur, scale change, and viewpoint change) with limited variations. Our study encompasses a wider variety of transformations and multiple variations per transformation, ensuring more comprehensive findings.	

Since all compared images were derived from the original images, a theoretical matching rate of 100% could be expected. However, upon examining Table 3, it becomes evident that some feature extractors failed to capture any matches at all. The highest occurrence of no matches was observed after the blurring process, followed by the rotation process. The most significant instance of no matches occurred when the STAR detector was paired with the FREAK descriptor, followed by the STAR detector with the BRIEF and DAISY descriptors. In contrast, when the SURF feature extractor was used both as a detector and descriptor, a matching result was obtained in nearly all images. This outcome was similarly observed with the SIFT and KAZE feature extractors.

The speed of detecting and computing keypoints in an image is another crucial factor in the selection of a feature extractor. Table 4 shows the average time spent on both feature detection and description processes, along with their standard deviation. As expected, the FAST algorithm was measured to be the fastest overall as a feature detector, with ORB following as the second fastest. In the feature description process, ORB, BRIEF, and FREAK were the fastest, all yielding approximately the same results. The KAZE algorithm consumed the most time. In conclusion, using FAST as the detector and ORB as the descriptor results in the fastest feature extraction process. Additionally, KAZE exhibited the highest standard deviation, indicating significant variability compared to other algorithms. This suggests that KAZE’s detection time is more inconsistent and differs more from other detection times, potentially making it an outlier in terms of performance. Furthermore, some of the maximum detector or descriptor times were quite high, which could be attributed to the operating system. Since the number of high values is relatively low and was measured instantaneously, the durations aligned more closely with the average value when the images with these high time durations were retested later. Therefore, comparing time duration by testing on a limited number of images is not appropriate. In this study, the average of measurements made on more than two million images has been used to exclude these high values.

Given that all the compared images originated from the original images, one might anticipate a theoretical matching rate of 100%. However, as shown in Table 5, the actual match results fall significantly short of this expectation. Additionally, the table indicates that the number of matched features with the kNN matcher is lower than with the BF Matcher. This discrepancy arises because the kNN matcher tends to produce a more focused set of matches, potentially reducing the number of incorrect matches by selecting only the k best matches (in this study, k is 2) for each feature. However, the kNN matcher requires more time for the matching process. With the BF matcher, the shortest time per feature for matching was measured using ORB, while the longest time was observed in BF matching of features obtained with SIFT. In contrast, with the kNN matcher, the shortest time was recorded for matching features obtained with STAR + BRIEF, whereas the longest time was measured in matching features obtained with FAST + ORB. Furthermore, the low standard deviations for both matching times indicate that the detection times are more consistent and similar across different executions. This suggests that these algorithms have stable performance in terms of detection time with minimal variation.

Figure 3 illustrates that the best results are achieved with STAR as the detector and DAISY as the descriptor up to 20 degrees of affine transformation. However, beyond 20 degrees, the ORB feature extractor yields the best matching results. It can be concluded that ORB is more robust than other feature extractors after 20 degrees of affine transformation. Figure 4 elucidates the robustness of feature extractors when exposed to image blurring. In the context of blurring, the best matching results were achieved with AKAZE, followed by KAZE and SURF, which demonstrated approximately the same matching success. In contrast, the lowest matching results were obtained using STAR as the detector and FAST as the descriptor.

Figure 5 illustrates the matching performance of the feature extractors when the image is subjected to barrel fisheye distortion. In response to this geometric distortion, the highest matching results were achieved using SURF, SIFT, KAZE, and AKAZE, with outcomes that were approximately similar. Conversely, the lowest level of matching success emerged when STAR was used as the detector and FAST as the descriptor.

Vertical and horizontal perspective distortions are common types of distortions encountered in image processing. Figure 6 illustrates the robustness of the feature extractors to these distortions. As shown in the figure, the matching success levels of the algorithms vary continuously depending on whether the distortion is horizontal or vertical and the degree of distortion. For vertical perspective distortion at 5 degrees, the best matching results were achieved with the STAR + DAISY, STAR + BRIEF, and AKAZE feature extractors with nearly identical values. When the distortion degree increased to 10 degrees, the best result was achieved with STAR + DAISY. As the distortion degree increased to 15 degrees, the best results were achieved with STAR + DAISY and AKAZE. When the distortion degree reached 20 degrees, the best results were obtained with AKAZE, STAR + DAISY, and ORB. At 25 degrees of distortion, the best result was achieved with the ORB feature extractor. Thus, it can be concluded that the most resilient methods against vertical perspective distortions are STAR + DAISY and AKAZE. Similar findings were observed in the context of horizontal perspective distortions, albeit with minor variations. For horizontal perspective distortion at 5 degrees, the best matching results were achieved with the AKAZE, STAR + DAISY, and STAR + BRIEF feature extractors with nearly identical values. When the distortion degree increased to 10 or 15 degrees, the best result was achieved with STAR + DAISY. As the degree of distortion increased to 20 degrees, the best results were achieved with STAR + DAISY, followed by AKAZE. At 25 degrees of distortion, the best result was achieved with the ORB feature extractor. Collectively, these findings suggest that the STAR + DAISY and AKAZE feature extractors offer the most robust solutions against horizontal perspective distortions.

In the domain of image processing, one of the particular challenges is the impact of illumination changes, such as brightening, on the matching accuracy of feature extractors. Figure 7 provides a comprehensive overview of how different feature extractors respond to the challenge of brightening in images. As the brightness level of the images increases, the matching success of all feature extractors decreases, aligning with theoretical predictions. However, despite this, the highest matching rate across all brightness levels was achieved with the ORB feature extractor. Furthermore, it has been observed that as brightness levels escalate, the disparity in matching performance among the various feature extractors diminishes, suggesting a convergence in their robustness to luminosity variations. Figure 7 also illustrates the comparative efficacy of the feature extractors against salt-and-pepper distortion, which was administered at divergent intensities. Salt-and-pepper noise is one of the most challenging distortions during the feature extraction and matching process, as it introduces sharp and random fluctuations in pixel values, significantly obscuring image features. Optimal performance was achieved by the BRISK algorithm, followed by the FAST + ORB algorithms. However, it is imperative to acknowledge that the presence of salt-and-pepper noise beyond a certain threshold may induce anomalously high matching rates even among images that lack identicality.

From a theoretical standpoint, one might expect that an image rotated from the original would remain identical to the original. Nevertheless, the matching results obtained by comparing the rotated images with the original image are presented in Fig. 8. The highest matching rate in the rotated images was achieved with AKAZE, followed by SIFT. Conversely, the minimal matching values were observed with STAR + DAISY, STAR + FAST, and STAR + BRIEF, all of which exhibited approximately the same values. The highest matching success with AKAZE was achieved between 160 and 200 degrees, indicating that the image is nearly completely flipped upside down.

Scaling, a common operation in image manipulation, involves resizing images either by enlarging or reducing them, which can significantly affect the detectability of features within these images. Figure 9 demonstrates the matching success of the feature extractors discussed in this study in response to scaling. As image sizes are reduced, the matching rates of all feature extractors decrease, converging towards each other. However, as images are enlarged, the matching rates achieved by some algorithms increase up to a certain point. The best results against scaling were achieved with AKAZE. Beyond an image enlargement rate of 220%, no significant increase in success rate was measured for AKAZE.

Conclusions

Feature extraction is of paramount importance in the domain of computer vision, serving as a cornerstone in the analysis, interpretation, and understanding of visual data. Central to the efficacy of numerous computer vision tasks, feature extractors play a pivotal role in identifying and capturing salient patterns, structures, and attributes within images. Selecting the most suitable feature extractor is crucial for achieving the desired objectives in machine vision. This study, through the execution of over two million comparisons, has revealed which feature extractor is most appropriate under varying conditions such as rotation, scaling, blurring, brightening, affine transformation, horizontal/perspective transformation, fisheye distortion, and salt-and-pepper noise. Key findings from this study reveal that: In scenarios where feature extraction and matching time are critical, such as in real-time applications, employing the FAST algorithm as the detector, the ORB algorithm as the descriptor, and BF matcher for the matching process will result in the fastest feature extraction and matching procedure. In situations where an affine-transformed image is expected, the ORB algorithm should be preferred.

AKAZE demonstrated superior robustness against blurring, rotation, and scaling, making it ideal for applications where image quality may be compromised.

In the event of working with images subjected to Barrel Fisheye Distortion, it is advisable to employ one of the algorithms—SURF, SIFT, KAZE, or AKAZE—which have demonstrated similar performance in terms of matching accuracy.

Should the anticipated task involve processing images with horizontal or vertical perspective distortions, it is recommended to utilize either the AKAZE algorithm or the STAR algorithm as the detector with the DAISY algorithm as the descriptor for optimal performance.

In scenarios characterized by significant variations in image brightness, the ORB algorithm should be preferred.

Should the images be expected to undergo substantial exposure to salt-and-pepper noise, the selection of the BRISK algorithm is advised for optimal performance.

The results of this comprehensive analysis underscore the importance of selecting the appropriate feature extractor based on the specific challenges and requirements of the application at hand. The insights gained from this study can guide the development of more robust and efficient computer vision systems, particularly in real-time or resource-constrained environments.

Future work could explore the integration of these traditional feature extractors with modern deep learning-based techniques to further enhance their performance. Additionally, the dataset used in this study could be expanded to include more diverse image types and conditions, providing a broader foundation for evaluating feature extractors in even more challenging scenarios.

While this study provides a comprehensive evaluation of traditional feature extractors, it is important to acknowledge certain limitations. One primary limitation is that the dataset, although extensive, may not fully capture the diversity of real-world conditions. The specific characteristics of the dataset, such as the types of transformations applied and the inherent properties of the images, may have influenced the outcomes of the evaluation. Another limitation is that the computational performance of the evaluated algorithms was measured on a single hardware configuration, limiting the generalizability of these results to other platforms. Variations in hardware, such as differences in CPU, GPU, or memory capabilities, could lead to different performance outcomes.

Additional Information and Declarations

Competing Interests

Author Contributions

Data Availability

The authors declare that they have no competing interests.

Murat ISIK conceived and designed the experiments, performed the experiments, analyzed the data, performed the computation work, prepared figures and/or tables, authored or reviewed drafts of the article, and approved the final draft.

The following information was supplied regarding data availability:

The reproduced dataset is available at Kaggle: https://doi.org/10.34740/kaggle/ds/4493370.

The Image Matching Challenge PhotoTourism (IMC-PT) 2020 dataset is available at: https://www.cs.ubc.ca/%7Ekmyi/imw2020/data.html.

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
