# Peer review of "Comprehensive empirical evaluation of feature extractors in computer vision"

_PeerJ Computer Science, doi:10.7717/peerj-cs.2415_

## Round 0.1 · original submission · Major Revisions

Dear authors,

Thank you for submitting your article. Based on reviews' comments, your article has not yet been recommended for publication in its current form. However, we encourage you to address the concerns and criticisms of the reviewer and to resubmit your article once you have updated it accordingly. Reviewer 2 and Reviewer 4 have asked you to provide specific references. You are welcome to add them if you think they are relevant. However, you are not obliged to include these citations, and if you do not, it will not affect my decision.

Beest wishes,

·

Basic reporting

1- Abbreviations were used recklessly without any initial definition. Please define all the keywords before abbreviating them.
2-Remove (we) you have many. Replace with, for example (This study) or (The proposed model).

Experimental design

1. Add Limitations of the proposed study need to be discussed before conclusion.
2. There are any Enhancement on your work? or you are applied the algorithms directly? Please clarify it. And what is the different between another works? Your comparison is not clear. If you are using the algorithms directly without any modification, so what is the main contribution in your work.

Validity of the findings

The related work section should provide more information about the differences between the cited solutions and the proposed paper, exposing the improvements introduced.

Analysis is required for each experiment to show its main purpose. Furthermore, the paper needs clear flow diagram for all proposed steps.

In table 6, author show results, how these results were generated. How author verify that his/her accuracy is best than other.

Need to put some simple of data in your study, and how did you deal with it.

Were there any potential limitations in the selection of the datasets used in the study, and how might this impact the generalizability of the findings to other applications?

Overall, the basic background is not introduced well, where the notations are not illustrated much clear. I recommend the authors to employ certain intuitive examples to elaborate the essential notations.

Additional comments

The novelty of this paper is not apparent, you should highlight your contribution in the end of introduction in detail.

·

Basic reporting

The paper is well written and organized. The authors should revise the manuscript taking all the following points into considerations.
1- Abstract should enhanced and include the main numerical results obtained.
2- Introduction is very lack you must include
- research gap
- research motivation
-research quaesion
- problem statement
-major contribution
- paper organization

Experimental design

The exeperimental results are nor sufficient
1- in line 252 Image Deriving Strategy
-253 We manually apply different types of transformations as listed below on original images and
-254 compute the matching evaluation parameters such as the count of keypoints, time duration of
-255 descripting / matching, and matching performance.
Please add explainable example with numerical results
2- In line 348 how your study utilized 4282 images, with a total of 196.972 variations. Please more images (samples are required)
3- Low resolution images please enhance.

Validity of the findings

You need to boost the findings with some statistical analysis.

Additional comments

I recommend the to expand more related works as follows
- Hassan, Esraa, Mahmoud Y. Shams, Noha A. Hikal, and Samir Elmougy. "A novel convolutional neural network model for malaria cell images classification." Computers, Materials & Continua 72, no. 3 (2022): 5889-5907.
- Hassan, E., Shams, M. Y., Hikal, N. A., & Elmougy, S. (2023). The effect of choosing optimizer algorithms to improve computer vision tasks: a comparative study. Multimedia Tools and Applications, 82(11), 16591-16633.

Reviewer 3 ·

Basic reporting

The langugae can be enahanced and espectially lterature survey can be snipped to suit the context.

Experimental design

reviewing the performance of 10 feature descriptor/detectors is appreciated. But authors need to establish the motivation & significance to chose these 10 algorithms.

Validity of the findings

There is no tile description for x and y axes. why authors used line plot when barplot is sufficient for figure-5.

Additional comments

I appreiate the interst and effort of the authors in conucting this research.

But following are few concerns:

1. Whats the motivation to choose these 10 feature extraction techniques?
2. In ln.No. 60-61, authors mentioned several parameters to compare the algorithms. Are authors considering all these parameters for comparision?
3.Literature survey can be snipped .
4. In figures, what x and y axis represent?
5. What the inference from figure-5? Why line plot is chosen as the data in x-axis represent different algorithms?

Cite this review as

Reviewer 4 ·

Basic reporting

The authors present comprehensive evaluation of feature extractors in Computer Vision. The study is interesting. In general, the main conclusions presented in the paper are supported by the figures and supporting text. However, to meet the journal quality standards, the following comments need to be addressed.
• Abstract: Should be improved and extended. The authors talk lot about the problem formulation, but novelty of the proposed model is missing. Also provided the general applicability of their model. Please be specific what are the main quantitative results to attract general audiences.
• The introduction can be improved. The authors should focus on extending the novelty of the current study. Emphasize should be given in improvement of the model (in quantitative sense) compared to existing state-of-the art models.
• More details about network architecture and complexity of the model should be provided.
• what about comparison of the result with current state-of-the art models? Did authors perform ablation study to compare with different models?
• What are the baseline models and benchmark results? The authors may compared the result with existing models evaluated with datasets
• Conclusion parts needs to be strengthened.
• Please provide a fair weakness and limitation of the model, and how it can be improved.
• Typographical errors: There are several minor grammatical errors and incorrect sentence structures. Please run this through a spell checker.
Discussions of relevant literature could be further enhanced, which can help better motivate the current study and link to the existing work. Authors should consider the following relevant recent works in the field of applying SOTA deep learning techniques to better motivate the usefulness of machine learning approaches, such as
see :- Neural Networks 2022 https://doi.org/10.1016/j.neunet.2022.05.024
-Adv. Eng. Informatics 2023, 56, 102007, https://doi.org/10.1016/j.aei.2023.102007
Neural, Comput & Applic (2022) https://doi.org/10.1007/s00521-021-06651-x
Hence they should be briefly discussed in the related work section.

Experimental design

see above

Validity of the findings

see above

Additional comments

see above

Cite this review as

---

## Round 0.2 · accepted · Accept

Dear authors,

Thank you for revising the manuscript. Two of the original reviewers did not respond to the invitation for reviewing the revision. Other two reviewers think that the necessary additions and arrangements have been appropriately performed and the article seems to be sufficiently improved. As such, the article is considered acceptable.

Best wishes,

·

Basic reporting

The author reply all my concerns and the paper can be accepted in it's current form.

Experimental design

All my concerns are answered

Validity of the findings

Good

Additional comments

No

Reviewer 4 ·

Basic reporting

The revised manuscript is now suitable for publication in Peer J.

Experimental design

NA

Validity of the findings

NA

Cite this review as